# Factors Impacting σ- and π-Hole Regions as Revealed by the Electrostatic Potential and Its Source Function Reconstruction: The Case of 4,4′-Bipyridine Derivatives

**DOI:** 10.3390/molecules25194409

**Published:** 2020-09-25

**Authors:** Carlo Gatti, Alessandro Dessì, Roberto Dallocchio, Victor Mamane, Sergio Cossu, Robin Weiss, Patrick Pale, Emmanuel Aubert, Paola Peluso

**Affiliations:** 1CNR-SCITEC, Istituto di Scienze e Tecnologie Chimiche “Giulio Natta”, sezione di via Golgi, via C. Golgi 19, 20133 Milano, Italy; 2Istituto Lombardo, Accademia di Scienze e Lettere, via Brera 28, 20100 Milano, Italy; 3Institute of Biomolecular Chemistry ICB, CNR, Secondary branch of Sassari, Traversa La Crucca 3, Regione Baldinca, Li Punti, 07100 Sassari, Italy; alessandro.dessi@cnr.it (A.D.); roberto.dallocchio@cnr.it (R.D.); 4Strasbourg Institute of Chemistry, UMR CNRS 7177, Team LASYROC, 1 rue Blaise Pascal, University of Strasbourg, 67008 Strasbourg, France; robin.weiss@unistra.fr (R.W.); ppale@unistra.fr (P.P.); 5Department of Molecular Science and Nanosystems DSMN, Venice Ca’ Foscari University, Via Torino 155, 30172 Mestre Venezia, Italy; cossu@unive.it; 6Crystallography, Magnetic Resonance and Modelling (CRM2), UMR CNRS 7036, University of Lorraine, Bd des Aiguillettes, 54506 Vandoeuvre-les-Nancy, France; emmanuel.aubert@univ-lorraine.fr

**Keywords:** atomic group contributions, bipyridines, chalcogen bond, electrostatic potential, halogen bond, σ-hole, π-hole, source function

## Abstract

Positive electrostatic potential (*V*) values are often associated with σ- and π-holes, regions of lower electron density which can interact with electron-rich sites to form noncovalent interactions. Factors impacting σ- and π-holes may thus be monitored in terms of the shape and values of the resulting *V*. Further precious insights into such factors are obtained through a rigorous decomposition of the *V* values in atomic or atomic group contributions, a task here achieved by extending the Bader–Gatti source function (SF) for the electron density to *V*. In this article, this general methodology is applied to a series of 4,4′-bipyridine derivatives containing atoms from Groups VI (S, Se) and VII (Cl, Br), and the pentafluorophenyl group acting as a π-hole. As these molecules are characterized by a certain degree of conformational freedom due to the possibility of rotation around the two C–Ch bonds, from two to four conformational motifs could be identified for each structure through conformational search. On this basis, the impact of chemical and conformational features on σ- and π-hole regions could be systematically evaluated by computing the *V* values on electron density isosurfaces (*V*_S_) and by comparing and dissecting in atomic/atomic group contributions the *V*_S_ maxima (*V*_S,max_) values calculated for different molecular patterns. The results of this study confirm that both chemical and conformational features may seriously impact σ- and π-hole regions and provide a clear analysis and a rationale of why and how this influence is realized. Hence, the proposed methodology might offer precious clues for designing changes in the σ- and π-hole regions, aimed at affecting their potential involvement in noncovalent interactions in a desired way.

## 1. Introduction

The electrostatic potential computed on a molecular electron density isosurface (*V*_S_) is beneficially used to assess electronic properties and interaction capability of specific atoms and sites [1,2]. *V*(**r**) is a real physical property which represents the electrostatic potential in a point **r** [3]. *V*(**r**) is generated by each nucleus in a system and by the system’s electron distribution, and is given by Equation (1):(1)$$V(r)\;=\;\sum_{A}\frac{Z_{A}}{R_{A}-r}-\int \frac{\rho \left(r\right)dr^{\prime }}{|\;r^{\prime }-r|\;}$$
where Z_A_ is the charge on nucleus A located at **R**_A_, and *ρ*(**r**) is the electron density distribution. The sign of *V*(**r**) is positive or negative if the effect of the nuclei (first positive term) or that of electrons (second negative term) is dominant, respectively. Significantly, the analysis of *V*_S_ values disclosed that *V* around bound atoms of Groups III–VIII is anisotropic due to the presence of regions of electron charge density depletion (σ-holes) which are able to interact with electron-rich regions [4,5,6,7]. Several studies have proven that computing *V* associated with a σ-hole in its unperturbed state can furnish a useful estimation of the strength of potential noncovalent interactions involving the electropositive region [2,5]. Among noncovalent interactions related to the σ-hole concept, the well-known halogen [8] and chalcogen [9] bonds (XB and ChB, respectively) have been studied and applied in several fields for decades [10,11,12,13]. Because *V* in a region depends on the contributions of the whole molecule [14,15], seminal studies have focused on the structural factors which affect local *V* by using simple compounds as test probes [14,16,17,18]. On the other hand, understanding the factors which impact σ-hole in complex systems is important in designing new molecules for applicative purposes [19,20,21,22]. Many XB and ChB donors reported so far are based on the use of electron-attracting heterocyclic systems which induce polarization in bound atoms of Groups VI and VII, enhancing their electrophilic character [12,23,24]. Fluorination also increases σ- and π-hole depth [25], π-holes being electron deficient regions often observed on polarized double bond and π-acidic aromatics, and able to interact with nucleophiles [26,27].

In the last few years, our groups have demonstrated that the 4,4′-bipyridyl moiety provides polarization of bound X atoms. Halogenated 4,4′-bipyridines have been found to act as XB donors in solid-state [28,29] and in solution [30,31]. Recently, the series of 5,5′-dibromo-2,2′-dichloro-3-chalcogeno-4,4′-bipyridines **1**–**6** (Figure 1) were designed and prepared as multi-site σ-hole donors [32,33], derivatives **3** and **6** also containing a π-hole centered on a pentafluorophenyl ring. In particular, compounds **2**, **3** and **6** were shown to function as σ- and π-hole donors in chiral recognition [32], ^19^F-NMR titration and catalysis experiments [33]. From the structural point of view, compounds **1**–**6** bear different combinations of Ch atom (S and Se) and a distinctive substituent on Ch (Me, Ph, C_6_F_5_) on a tetrahalogenated 4,4′-bipyridine scaffold (X = Cl, Br). Moreover, the 3-chalcogeno-4,4′-bipyridyl motif is characterized by a degree of conformational freedom due to the possibility of rotation around the two C–Ch bonds (Figure 1). On the contrary, rotation around the 4,4′-axis is restricted due to the effect of sterically hindered 3,5,5′-substituents (atropisomerism), and consequently the two pyridyl rings (*pyr* and *pyr’*) are not coplanar and almost orthogonal. In general, motifs used as ChB donors for applicative purposes are characterized by low conformational flexibility to enhance the directionality feature of the noncovalent interaction [12,34]. On the other hand, to date systematic studies and observations focused on the impact of molecular conformational features on σ- and π-holes incorporated in a structure are scarcely reported [35,36,37,38].

In this article, using compounds **1**–**6** as test probes to explore a multi-site system, we describe a study aimed at evaluating how the impact of chemical and conformational features on the environment around σ- and π-holes affects the *V*_S_ maxima (*V*_S,max_) related to these regions. Our study is composed of two steps in sequence (Section 2). In the first one (Section 2.1 and Section 2.2), we search the stable conformers of each compound, we calculate their corresponding *V*_S,max_ values and analyze the factors impacting σ- and π-holes in terms of the shape and values of the resulting electrostatic potentials. Then (Section 2.3 and Section 2.4), a rigorous decomposition of the *V*_S,max_ values in atomic or atomic group contributions is afforded, by extending the Bader–Gatti source function (SF) for the electron density [32,39,40,41] to *V* (which is analogous to adopt a suitable atomic partition for *ρ* (**r**) in Equation (1)). Factors impacting σ- and π-holes may be so dissected in separate contributions from chemically meaningful moieties of the molecule that regardless of their being close or far from the holes, can either (over/partly) contribute to or oppose the observed *V*_S,max_ values. Such a dissection provides unprecedented insights into the factors leading to *V*_S,max_ value changes and trends. In fact, as an example, while it is well acknowledged that σ-holes originate from the cylindrical symmetry of the σ-bond and the more or less asymmetric electron sharing along its axis, nothing is quantitatively known about the synergic or antagonist roles played by the various moieties of a molecule in producing such holes. Section 3 reports technical details about our adopted methodology and the performed computations, while Section 4 concludes.

It is expected that our two-step methodology may serve as a valid tool for designing changes in the σ- and π-hole regions of families of molecular compounds, aimed at affecting their (potential) involvement in noncovalent interactions in a desired manner.

## 2. Results and Discussion

In a previous study, we demonstrated that different conformers of compounds **3**–**6** exist in ethanol [33]. On this basis, with the aim of identifying low energy conformers for compounds **1**–**6** in vacuum, a conformational search procedure was carried out for each compound (see Appendix A for energies, geometric parameters, and Boltzmann distributions). Conformational motifs A1 and A2 were found for compounds **1**, **4**, and **5**, and A1, B1, A2, and B2 for compounds **2**, **3**, and **6** (Figure 2 and Appendix A).

They originate from the relative orientation of the 3 substituents, the methyl group (for **1** and **4**), the phenyl (for **2** and **5**), and the pentafluorophenyl (for **3** and **6**) rings. These substituents can be in front (conformation A) of the 2′-chloro-5′-bromo-4′-pyridyl ring (*pyr’*) or away from it (conformation B) due to rotation around the bond C3–Ch. For each of the two conformations A and B, two additional conformers are generated by the relative position of the 3 substituents, which can be close to the 3′-hydrogen (conformations A1 and B1) or to the 5′-bromine atom (conformations A2 and B2) of the *pyr’* ring. It is worth noting that XBs and a π-hole bond were observed for conformer **6**-B2 [33] by X-ray diffraction analysis. As depicted in Appendix A, during this study a Se···N contact (ChB) was also detected in the crystal packing of conformer **5**-B2.

### 2.1. Calculation of V_S,max_ Values

Using the 18 conformers of compounds **1**–**6** as test probes, we computed *V*_S,max_ values for 2-Cl, 2′-Cl, 5-Br, 5′-Br, 3-S (compounds **1**–**3**) and 3-Se (compounds **4**–**6**) σ-holes, and phenyl π-holes (compounds **2**, **3**, **5** and **6**) (Table 1). Recently, for compounds containing Se and X atoms, Murray and co-workers demonstrated the need for polarization functions when computing *V* [17].

A comparative analysis performed by using conformer **6**-B1 as a representative case confirmed that for basis sets such as 3-21G* and larger, results and trends of *V*_S_ values associated with the holes for the B3LYP and M06-2X methods do not greatly differ (see Appendix A). On this basis, the calculations were performed in vacuum, at DFT level of theory, with the B3LYP functional and 6-311G* as basis set mapping the *V*_S_ on a 0.002 au isodensity surface of the unperturbed molecules [2]. This isodensity surface was used to detect properly small σ-holes. Indeed, using conformer **1**-A2 as a model, the *V*_S,max_ on the sulfur σ-holes were computed, by using different isodensity envelopes ranging from 0.001 to 0.004 au (Table 2). The surface of bonded sulfur along the C–S bond elongations does not enclose a σ-hole when the potential is mapped on the 0.001 au isodensity surface.

However, choosing 0.002 au as isodensity surface value provides the appearance of *V*_S,max_ on sulfur along the C_pyridyl_–S bond extension, whereas the σ-hole along the extension of the C_Me_–S bond is only observed by using the closer 0.004 au isosurface. This comparison shows that the 0.001 au isosurface is not suitable for computation of *V*_S,max_ on regions of electron charge density depletion located on less polarizable atoms such as sulfur [42]. It is worth mentioning that the 0.002 au isodensity surface has been also preferred with respect to the 0.001 au one in some studies [42,43,44].

### 2.2. Impact of Chemical and Conformational Features on V_S_ Values

Each conformer is characterized by three variable features: (i) the Ch atom (S or Se), (ii) the R group on Ch (Me, Ph or C_6_F_5_), (iii) the conformational motif (A1, A2, B1, or B2). On the contrary, all conformers contain the 5,5′-dibromo-2,2′-dichloro-4,4′-bipyridyl as fixed motif. The impact of the structural (chemical + conformational) features on σ- and π-holes was estimated by calculating for each hole the variation range of *V*_S_,_max_ (Δ*V*_S,max_) (Table 1). It was found to increase following the order 2′-Cl < 5-Br < 5′-Br < 2-Cl < 3-Ch (C_pyr_–Ch bond) < 3-Ch (C_R_–Ch bond) < π-hole.

#### 2.2.1. σ-Holes Located on the Elongation of C-X Bonds (X = Cl, Br)

As expected, the *V*_S_,_max_ values are more positive for bromine σ-holes compared to chlorine ones due to the higher polarizability and lower electronegativity of Br (3.05 × 10^−24^ cm^3^ and 2.96, respectively vs. 2.18 × 10^−24^ cm^3^ and 3.16 for Cl) [45,46]. The *V*_S_,_max_ variation for halogen σ-holes as the R group changes follows the order Ph < Me < C_6_F_5_ in almost all cases. In general, more positive *V*_S_,_max_ values for halogen σ-holes were found in compounds **1**–**3** compared to **4**–**6**. This trend reflects the higher electron-attracting power of sulfur compared to selenium, the Pauling electronegativity [45] of sulfur (2.58) being slightly higher than that of selenium (2.55). The effect of Ch electronegativity makes *V*_S_,_max_ more positive on chlorine and bromine σ-holes which are located on the same ring of the Ch atom (*pyr* ring) with respect to the two halogen σ-holes on the other ring (*pyr’* ring). Some observed deviations from this trend are likely due to through-space effects related to the position of positive and negative regions on the 3 substituents as the conformational motif changes. In fact, for the σ-holes located at the elongations of the C2–Cl and C5′–Br bonds, the *V*_S_,_max_ values could be affected by the proximity of either the R group (conformers A2 and B2) or the σ-hole on the elongation of the C_R_–Ch bonds (conformers A1 and B1). In contrast, *V*_S_,_max_ of σ-holes located at the elongations of the C2′–Cl (Δ*V*_S_,_max_ = 0.0046 au) and C5–Br (Δ*V*_S_,_max_ = 0.0055 au) bonds show to be less affected by structural changes due to the fact that their environment remains almost unchanged as chemical and conformational features change.

#### 2.2.2. Phenyl π-Holes

The π-holes on the phenyl ring of compounds **2**, **3**, **5**, and **6** are deeply affected by the electron-attracting power of the substituents on the phenyl ring (C_6_H_5_ vs C_6_F_5_). Indeed, positive *V*_S_,_max_ are only observed on the pentafluoro-substituted phenyl ring (compounds **3** and **6**), whereas negative values were computed for the phenyl ring in **2** and **5**. Following the electronegativity scale, the sulfur atom at the 3-position in compound **3** exerts a slightly stronger electron-attracting effect on the pentafluorophenyl ring compared to selenium atom in compound **6**. Accordingly, the π-hole *V*_S_,_max_ in compound **3** (average values 0.0461 au) is just slightly more positive than the π-hole *V*_S_,_max_ observed in compound **6** (average values 0.0448 au). For comparison purpose, we computed the π-hole *V*_S_,_max_ of the 5,5′-dibromo-2-2′-dichloro-3-(perfluorophenyl)methyl-4,4′-bipyridine (**7**, see Appendix A for details) which bears a CH_2_ unit in place of the Ch atom. For **7**, the π-hole *V*_S_,_max_ becomes less positive (average values 0.0401 au) with respect to the *V*_S,max_ computed for the (perfluorophenyl)selanyl derivative **6** even if the same electronegativity is given for both Se and C (2.55). This trend could be rationalized in terms of atomic polarizabilities, selenium being more polarizable (3.77 × 10^−24^ cm^3^) than carbon (1.76 × 10^−24^ cm^3^) [46], and for this reason more prone to accommodate the gain of electron charge density [16].

*V*_S_,_max_ regions are found above and below the planar phenyl rings in almost all cases. In particular, the π-hole regions are located on the phenyl face far from (external) and close to (internal) the 4,4′-bipyridine scaffold, respectively. These two *V*_S_,_max_ regions are not equal because the face close to the bipyridine framework (internal) is more affected by the features of different conformational motifs. Consequently, considering compounds **3** and **6**, the *V*_S_,_max_ of the external region varies over a short range (0.0018 and 0.0017 au, respectively), whereas the *V*_S_,_max_ of the internal region covers a wider range (0.0134 and 0.0137 au, respectively). In particular, two extreme situations can be observed. Indeed, in conformers **3**-A2 and **6**-A2 no *V*_S_,_max_ was found in the internal side of the aromatic ring. This phenomenon may be due to the overlapping of the negative *V* located on the side of the bromine at position 5′. In this case, the *V*_S,max_ value is affected by a through-space effect [15] determined by an intramolecular Br···π-hole contact. In contrast, for compounds **3** and **6** the highest π-hole *V*_S_,_max_ values are observed for conformers A1 and B1 because the phenyl ring is far from the 5′-bromine and, consequently, does not undergo any saturation effect.

#### 2.2.3. 3-Ch σ-Holes

In principle, in compounds **1**–**6**, the Ch atoms can bear two σ-holes on the elongation of the C_pyridyl_–Ch and C_R_–Ch bonds. In general, the main factors affecting the σ-hole *V* of a given R-Ch-R’ system are electronegativity and polarizability of Ch as well as the electron-attracting power and the polarizability of R and R’ [16]. In all considered conformers, the *V*_S_,_max_ values are more positive for selanyl compounds **4**–**6** compared to the thio-substituted series **1**–**3**. This trend is due to the higher polarizability (3.77 × 10^−24^ cm^3^) [46] and lower electronegativity (2.55) of selenium with respect to sulfur (2.90 × 10^−24^ cm^3^; 2.58). For the σ-hole located on the elongation of the C_pyridyl_–Ch, *V*_S_,_max_ values vary following the order Ph < Me < C_6_F_5_, whereas high variability is observed for the other σ-hole. The analysis of the *V*_S_,_max_ reported in Table 1 shows that in conformers A1 and B1 of compounds **2**, **3**, **5**, and **6**, *V*_S,max_ values on Ch are more positive for the C_pyridyl_–Ch σ-holes compared to the other σ-holes located on the elongation of the C_R_–Ch bonds. The opposite situations occur for conformers A2 and B2. Therefore, based on the trend observed for A2 and B2, the electron-attracting power (or/and polarizability) of the aryl rings is higher than that of the 4,4′-bipyridyl scaffold. Thus, in A1 and B1, a decrease of *V*_S,max_ occurs for the σ-hole located on the elongation of the C_R_–Ch bonds. Considering **6**-B1 and **6**-B2 as representative cases (Figure 3), the question seems to be related to the impact of the conformation environment on the C_R_–Se σ-hole. In fact, in **6**-B1 (Figure 3, left), the C_ArF_–Se σ-hole is oriented toward the 5′-bromine atom which contributes to decrease the *V*_S_,_max_ of the hole by partially saturating the neighboring positive region through the negative electrostatic potential located on the halogen side. This effect does not occur in conformer **6**-B2 (Figure 3, right) where the C_ArF_–Se σ-hole is oriented toward the small 3′-hydrogen atom. Interestingly, the unusual presence of two σ-holes (Figure 4, M1 and M2) on the elongation of the C_ArF_–Ch bond in conformers **3**-B1 and **6**-B1 appeared to be related to the features of the B1 motif. As mentioned above, in this conformation the σ-holes on the elongation of C_ArF_–Ch bond points towards the cylindrical region of negative *V* associated with the π-electron distribution along C–Br bond (the C5′-Br-M1(M2) angle is not far from 90°) (Figure 4 and Table 3). This makes the *V*_S_,_max_ of C_ArF_–Ch σ-hole partly counterbalanced by a region of minimal *V*, hence the maxima are forced to displace and split in two maxima M1 and M2. The negative *V* contribution from the neighboring bromine atom decreases the *V*_S_,_max_ values on C_ArF_–Ch σ-hole in **3**- and **6**-B1 compared to **3**- and **6**-B2. The σ-hole splitting phenomenon is rare but not unknown and it was observed in hypervalent iodine [47]. It is interesting to note that as a matter of fact, in conformers **3**-B1 and **6**-B1, 5′-bromine atom exerts a neighboring group stabilization by Ch σ-hole [35]. Accordingly, for both **3** and **6**, the B1motif corresponds to the lowest energy conformers in vacuum.

A different trend is observed for compounds **1** and **4**, likely due to the poor electron-attracting power of the methyl group. In particular, for **1**-A2 no σ-hole on the elongation of C_Me_–S bond was detectable on the 0.002 isosurface (Table 2). Analogously, no σ-hole was found in **2**-B2 on the elongation of C_Ph_–S bond.

### 2.3. Source Function (SF) Reconstruction of V_S,max_

Each *V*_S_,_max_ value for systems **1**–**6** may be decomposed as *V*_S_,_max_ = SF(Ch) + SF(R) + SF(Bipy) + Err(*V*_S_,_max_), where SF(Y) (Y = Ch, R, Bipy) denotes the cumulative SF contribution of the Y moiety, and Err(*V*_S_,_max_) the numerical integration error associated with the *V*_S_,_max_ SF reconstruction. *V*_S_,_max_ values and their SF(Y) contributions to the C_pyridyl_–Ch and the C_R_–Ch (Ch = S, Se) σ-holes and to the aryl π-holes, calculated for the various conformers of systems **1**–**6**, are reported in Table 4 and Table 5 (σ-holes) and S5 (π-holes). For the sake of clarity, two examples of such SF reconstructions for the C_pyridyl_–Ch σ-holes and other two for the aryl π-holes are illustrated in Figure 5 and Figure 6, respectively. The SF(Y) values were obtained through the ANASFR_EP code (see Section 3) from the atomic SF data. SF data for 3′-H and for 5′-Br atoms are listed in Table 4 and Table 5, and Appendix A due to the special role the atoms at these positions play, according to the various molecular conformations. In addition, Table 4 and Table 5, and Appendix A report the dissection of the SF(Bipy) contribution in its two composing ring contributions, SF(*pyr*) and SF(*pyr’*) and the SF integration errors. The latter are listed as percentage SF reconstruction errors for each *V*_S_,_max_ value, Err%(*V*_S,max_) = [(SF(Ch + R + Bipy) − *V*_S,max_)/*V*_S,max_] × 100.

#### 2.3.1. C_pyridyl_–Ch σ-Holes

With reference to the decomposition of such hole potentials (Table 4) into Ch, R and Bipy SF contributions, one notices that the *V*_S_,_max_ values are largely dominated by the SF(Ch) positive contribution. In particular, the one due to Se is about twice as large as that due to the S atom, for a given R and conformation. The SF(Ch) contribution is in most cases even larger than the *V*_S_,_max_ value and ranges, in percentage, from 76% to 134% for Ch = S, and from 169% to 234% for Ch = Se, depending on the conformation. This roughly doubled effect (in absolute value and in percentage) of the Se is not surprising, due to the lower electronegativity and thus the larger positive charge of Se relative to S (see infra). Conversely, this same reason increases the electronic charge on the R group. It so lowers the SF(R) contribution, changing it from moderately positive (**1**-A1, **1**-A2) to moderately negative (**4**-A1, **4**-A2) or from negligibly negative (**2**-A1, **2**-A2, **2**-B2) or moderately negative (**3**, all conformers) to largely negative (**5** and **6**, all conformers) on passing from Ch = S to Ch = Se. The trend SF(Me) > SF(Ph) > SF(C_6_F_5_) holds true for both Ch = S and Ch= Se and mirrors the trends of the Bader’s net charges q of the R group or, though understandably reversed, the trend of the positive Bader’s net charge on the Ch atom. For instance, for the A1 conformers, the q(R) charges decrease from 0.055 (Me), through −0.001 (Ph), to −0.149 (C_6_F_5_) for Ch = S and from −0.038 (Me), through −0.088 (Ph), to −0.251 (C6F5), for Ch = Se, while the q(Ch) charges increase from 0.091 (**1**-A1), through 0.122 (**2**-A1), to 0.209 (**3**-A1) for Ch = S and from 0.278 (**4**), through 0.315 (**5**), to 0.410 (**6**) for Ch = Se.

Analogously to the SF(R) contribution, also the SF contribution of the Bipy moiety decreases with the decrease of Ch electronegativity. For Ch = Se and for a given R, SF(Bipy) is always negative and greater in magnitude relative to Ch = S. For a given chalcogen, the SF%(Bipy) values are similar for R = Me or Ph (ranging from −14% to −20% for Ch = S and from −46% to –54% for Ch = Se), while the much larger electron-withdrawing power of the R = C_6_F_5_ group, shifts SF%(Bipy) to slightly positive values (4% to 9%) or to less negative values (about −26%) for Ch = S and Ch= Se, respectively. Although the positive SF(Ch) contributions always dominate the other two SF contributions to the *V*_S,max_ values, the relative magnitude of the generally negative (or seldom negligibly positive) SF(Bipy) and SF(R) contributions obviously depends on the R group and on the Ch type.

In particular, for Ch = S, the SF (R, R = Me or C_6_F_5_) value is from 2 to 10 times larger in magnitude than the SF(Bipy) value, while for R = Ph both contributions are small. However, SF(Bipy) is larger in magnitude than SF(Ph), which is negligibly negative or negligibly positive, depending on the conformer. Conversely, for Ch = Se, the SF(R) and SF(Bipy) contributions are more alike to each other in magnitude and always negative, hence they are always opposing the SF(Se) contribution. For Ch = Se, SF(Bipy) is twice as large in magnitude than SF(Me), while it is 2/3 (**5**, all conformers) or 1/3 (**6**, all conformers) smaller in magnitude than SF(R) for R = Ph or C_6_F_5_, respectively. Dissection of the Bipy source in the contribution from the *pyr* and *pyr’* ring moieties provides further insight. Regardless of the R group, of the system conformation and of the nature of Ch, the source from the *pyr* ring carrying the chalcogen atom is always negative, while that of *pyr’* ring is invariably positive. The magnitude of the source from the *pyr* ring significantly increases on passing from S to Se, becoming largely negative and bounded to a limited range of values (−0.020/−0.028 au), while that from the *pyr’* ring remains relatively small and in the range of 0.0035/0.0064 au for both Ch = S and Ch = Se. The different sign of the two ring sources, along with the large impact that the chalcogen nature has on the negative source of the *pyr* ring is a clear indication of the electron-donor effect of the chalcogen substituent on such ring. The SF contributions to the C_pyridyl_–Ch σ-holes from the 3′-H and 5′-Br atoms are almost constant for all **1**–**6** conformers, confirming that for such holes, due to their spatial location and at variance with the case of the C_R_–Ch σ-holes, the molecular conformation does not play any role. We conclude the discussion of Table 4 by noting that the percentage integration errors, Err%, are not negligible and occasionally large (3–4%) but generally lower than 1–2%. A similar comment holds true also for most of holes in Table 5 and Appendix A. Yet, the SF reconstruction of the phenyl π-holes with negative and small in magnitude (<0.01 au) *V*_S,max_ values, is by far less accurate (Err%: 3–13%).

#### 2.3.2. C_R_–Ch σ-Holes

Likewise the C_pyridyl_–Ch σ-holes, also the *V*_S,max_ values of the C_R_–Ch σ-holes (Table 5) are largely dominated by the positive SF(Ch) contribution, with those of Se being on average roughly twice as big as those from the S in magnitude. Although the SF(Ch) percentage contributions for the C_R_–Ch σ-holes are scattered over larger ranges of values (SF(S)%, 95–203; SF(Se)%, 175–317), compared to the case of the C_pyridyl_–Ch σ-holes (SF(S)%, 80–134; SF(Se)%, 169–235), the magnitudes of SF contributions do follow similar qualitative trends (C_R_–Ch σ-holes: **1** and **2**, 0.03/0.04 au, **3**, 0.05/0.06 au, **4** and **5**, 0.08/0.09 au, **6**, 0.08/0.12 au; C_pyridyl_–Ch σ-holes: **1** and **2**, 0.03/0.04 au, **3**, 0.06/0.07 au; **4** and **5**, 0.09 au, **6**, 0.12 au; note that for **3**-B1, with two C_R_–Ch σ-holes, the hole with the higher *V*_S,max_ value was here considered). Given these similarities between the SF(Ch) values and trends of the C_R_–Ch and C_pyridyl_–Ch σ-holes, analogous interpretations to those put forward for Table 4 also hold for the case of C_R_–Ch holes in Table 5. The cumulative source of the Bipy and the R moieties generally opposes the source from the Ch atom, apart one single case (**3**-B2) where it slightly (7%) concurs to the overall *V*_S,max_ value of the C_R_–Ch σ-hole. The way they oppose is very much dependent on the system and on its conformation. It may be just a small opposition as for systems **2**-A2 and **3**-A2 (−11/−20%) or definitely a far larger one as for the two **6**-B1 holes (−171/−216%) or the **4**-A1 and **5**-A1 holes (−198/−162%). The Bipy and the R groups may either cooperate or oppose each other in their action as the ratio of their contributing sources ranges from −6.3 to 2.5, depending on systems and their conformers. Despite such a variety of behaviors, there are some general trends worth to be mentioned. In the case of Ch = Se, the SF contribution for both Bipy and R moieties is invariably negative (as it is for the C_pyridyl_–Ch σ-holes), with that of the Bipy generally and significantly prevailing in magnitude over the Me and Ph for systems **4** and **5.** Instead, when R = C_6_F_5,_ it is the negative source of this group which is larger in magnitude in three out of the five reported cases. Analogously to the case of the C_pyridyl_–Ch σ-holes, the SF(R) values follow the trend SF(Me) > SF (Ph) > SF(C_6_F_5_) for both Ch = Se and S, due to the corresponding increase of the net negative charge of the R group along the series. For Ch = S, SF(R) is positive (R = Me) or still positive but smaller (R = Ph), while it is negative (−19/−59%) for R= C_6_F_5_. Regarding the distinct contributions from the two rings of the Bipy moiety, some of the considerations reported for the C_pyridyl_–Ch σ-holes apply. In particular, i) the observation about the constantly negative source from the *pyr* ring and the always positive contribution from the *pyr’* ring, and ii) the large increase of the magnitude of the SF(*pyr*) on passing from Ch = S to Ch = Se, yet to values (−0.039/−0.532) which are almost twice as big as for the C_pyridyl_–Ch σ-holes. It is thus once more evident that the contribution of the Bipy moiety is the result of quite distinct sources from its two composing ring moieties. As anticipated, sources from the 3′-H and the 5′-Br atoms to the C_R_–Ch σ-holes potential maxima are significantly affected in their magnitudes by molecular conformation.

In particular, SF(5′-Br) is quite high in magnitude and negative for A1 (−0.009/−0.017 au) and B1 conformers (−0.017/−0.020 au; the higher magnitude value is reported for cases with two C_R_–Ch σ-holes maxima), while it is marginally negative or even positive for other conformers (A2 and B2). To be facing (A1 and B1) or not facing (A2 and B2) the σ-hole, drastically changes the role the 5′-Br atom plays in determining the corresponding *V*_S,max_ value. The source from 3′-H is invariably positive and large (0.010/0.026 au), but particularly large for those conformers where SF(5′-Br) is almost negligible. For instance, SF(3′-H) is as large as 0.025/0.026 au for the B2 conformers.

#### 2.3.3. Aryl π-Holes

Decomposing π-hole potentials into Ch, R and Bipy SF contributions (Appendix A) neatly highlights the reasons leading, for R = Ph, to negative and small in magnitude *V*_S,max_ values and, for R = C_6_F_5_, to large and positive *V*_S,max_ values, regardless of Ch being S or Se. When R = Ph (compounds **2** and **5**), the negative SF(R) contribution is roughly of the same order of magnitude as the positive SF(Ch) contribution and the two sources almost cancel each other. As a result, *V*_S,max_ turns out to be mostly determined by the negative and small SF(Bipy) contribution. Conversely, when R = C_6_F_5_ (compounds **3** and **6**), SF(R) becomes much smaller in magnitude than SF(Ch). In fact, the former source drastically decreases its magnitude relative to that observed for compounds **2** and **5**, while the latter source slightly increases its positive value relative to these compounds because C_6_F_5_ withdraws more electrons from the Ch atom than the Ph group does. Combined with the fact that also SF(Bipy) of **3** and **6** becomes smaller in magnitude than for compounds **2** and **5** and definitely smaller than SF(Ch), the largely positive SF(Ch) source provides about 90% of the *V*_S,max_ value for conformers **3** or even overdetermines largely (140–190%) this value for conformers **6**. The fact that for conformers **6** SF(Bipy) is (much) less negative than for compounds **2** and **5** or that it becomes close to zero, or even positive for conformers **3**, is easily explained. Indeed, the strong electron-withdrawing power of C_6_F_5_ dampens the electron flow from the Ch atom to the Bipy moiety making this moiety less negatively charged and with a correspondingly (much) less negative source, relative to the case of R = Ph [for instance, for A1 conformers, the Bipy net charge decreases in magnitude from −0.121 to −0.060 e^−^ (**2**→**3**) and from −0.227 to −0.159 e^−^ (**5**→**6**)]. Dissection of SF(Bipy) in its two ring components SF(*pyr*) and SF(*pyr’*) shows that the former is negative, as its six-membered-ring (6MR) carries the Ch substitution, and, except for **3**-A1, always prevails in magnitude, while the latter is always positive. The SF(Ph) for the π-hole potentials is always negative, because the negative contribution from the 6MR C atoms overweighs the positive SF contribution from their linked H atoms. On the other hand, for R = C_6_F_5_ the 6MR C atoms significantly increase their positive charge because of the H→F substitution and so provide an overall positive source contribution at the π-holes. When Ch = S, the negative SF contribution from the fluorine atoms prevails over the positive contribution from the 6MR, while just the opposite occurs when Se replaces S, because the C atoms become even more positively charged and their positive SF contribution significantly increases.

External and internal π-hole potentials exhibit qualitatively the same SF contribution trends as a function of the R and Ch nature. There is, however, an important exception for conformers **2**-A2 and **5**-A2, whose internal π-holes exhibit significantly more negative *V*_S,max_ values (−0.0138 and −0.0160 au) than those of their external π-holes (−0.0090 and −0.0081 au). Please note that the potential values for the internal π-holes are the more negative ones for the whole set of investigated systems. The apparent anomaly (illustrated in Figure 6 for the **5**-A2 π-holes) finds an easy explanation in terms of SF(5′-Br) values and considering that in the **2**-A2 and **5**-A2 conformers the 5′-Br atom points its π-cloud towards the phenyl ring internal π-hole. Indeed, the SF(5′-Br) value is always negative for these conformers but as small as −0.0023 and −0.0025 au for the external and about 6 times as large (−0.0153 and −0.0173 au) for the internal **2**-A2 and **5**-A2 π-holes. Such an increase in magnitude qualitatively explains the potential differences between the external and internal π-holes for these two conformers.

### 2.4. Analysis of V_S,max_ and Source Function (SF) Contributions Changes

Further precious insights on the trends of the *V*_S,max_ values and their atomic (group) sources may be gained by decomposing the changes in their values upon chemical substitution into chemically meaningful contributions. Appendix A displays *V*_S,max_ variations (Δ*V*_S,max_) and corresponding SF contributions changes (ΔSF) upon substitution of either Ch or R atom/group or of both of them for a given conformer of systems **1**–**6**. Δ*V*_S,max_ values and their composing ΔSF contributions for a system change X→Y are evaluated as ΔZ = Z(Y)-Z(X) where Z = *V*_S,max_ or SF. Conformers were kept fixed for any considered X→Y change avoiding to introduce a further (and possibly second-order) variable in the analysis. The Δ*V*_S,max_ values may be decomposed as
Δ*V*_S,max_ (X→Y) ≅ SF(Ch) + ΔSF(R) + ΔSF(Bipy)(2)
where the “≅” sign in Equation (2) accounts for the unavoidable numerical integration error in the ΔSF reconstruction of the Δ*V*_S,max_ value. In Appendix A, those ΔSF and ΔSF% values that refer to Ch and R moieties undergoing a substitutional change upon the X→Y change of system are shown in bold. Equation (2) may also be recast as Equation (3):
Δ*V*_S,max_ (X→Y) ≅ SF_substitution_ + ΔSF_rearrangement_(3)
where the ΔSF_substitution_ term includes all the sources of groups undergoing a change of chemical composition (shown in bold in Appendix A) while the ΔSF_rearrangement_ term is the sum of ΔSF (Bipy) and of those ΔSF(Ch) or ΔSF(R) contributions arising from the Ch and R moieties that are not substituted in the X→Y process (all sources yielding as a sum ΔSF_rearrangement_ are shown not in bold in Appendix A). For the sake of clarity, Figure 7 (left plot) illustrates the ΔSF decompositions given by Equations (2) and (3) in an exemplar case.

Both ΔSF_substitution_ and ΔSF_rearrangement_ may be large and often (much) larger in magnitude than the Δ*V*_S,max_ value they reconstruct (Appendix A). When larger in magnitude than Δ*V*_S,max_, these two additive terms need to oppose each other to yield the observed Δ*V*_S,max_ value (within the SF reconstruction error). Thus, a chemical moiety substitution quite often induces a substantial charge rearrangement also in those moieties whose chemical formula remains fixed. Such a moiety rearrangement is mirrored in its SF contribution change following the X→Y process. Said in other words, each Δ*V*_S,max_ value for a system change X→Y results from both the substituted and the only apparently untouched moieties. For the sake of conciseness, the analysis worked out in Appendix A refers only to the C_pyridyl_–Ch and the C_R_–Ch σ-holes.

#### 2.4.1. S**→**Se Substitutions

Looking at Appendix A in more detail, one observes that when R is fixed, ΔSF(Ch) for the C_pyridyl_–Ch σ-holes remains positive and almost constant (0.0552/0.0582 au) for all S**→**Se changes, while a larger spread (0.0324/0.0744 au) is observed for the C_R_–Ch holes. A similar behavior is found for ΔSF(R) and ΔSF(Bipy) which are however negative and thus opposing rather than contributing to the positive Δ*V*_S,max_ values. In particular, ΔSF(R) and ΔSF(Bipy) amount to −0.0281/−0.0325 au and −0.0163/−0.0186 au, respectively, for the C_pyridyl_–Ch σ-holes and to −0.0164/−0.0177 au and −0.0124/−0.0462 au, respectively, for the C_R_–Ch holes. Since ΔSF(Ch) values are all largely positive and fairly exceeding the corresponding Δ*V*_S,max_ values, ΔSF_rearrangement_ turns out to be also largely negative, almost constant for C_pyridyl_–Ch holes (−0.0453/−0.0514 au) and significantly more spread out (−0.0291/−0.0629 au) for the C_R_–Ch holes.

#### 2.4.2. R**→**R’ Substitutions

At variance with the case of fixed R and S**→**Se substitution, keeping the Ch fixed and changing R to R’ leads to more complex variations. On passing from R = Me to R’ = Ph or R’ = C_6_F_5_ or from R = Ph to R’ = C_6_F_5_, ΔSF(R→R’) is always negative. It is also largely spread out (−0.0069/−0.0371 au) and in all cases, save two from R = Me to R’ = Ph (**1**-A1→**2**-A1 and **1**-A2→**2**-A2, C_pyridyl_–Ch σ-holes), significantly opposing the potential change Δ*V*_S,max_. Conversely, ΔSF(Bipy) is always positive and largely varying (0.0033/0.0266 au). ΔSF(Ch) is also usually positive and quite large (0.0104/0.0315 au), despite being only due to charge density rearrangements within the S or Se atomic basins (plus changes due to variations in the σ-holes maxima locations). In three specific cases, however, ΔSF(Ch) is negligibly small (**1**-A1→**2**-A1; **1**-A2→**2**-A2; C_pyridyl_–Ch σ-holes) or even negative (**2**-B1→**3**-B1, C_R_–Ch lower magnitude σ-hole). ΔSF_rearrangement_ is always positive, generally quite large (0.0144−0.0430 au) and significantly overdetermining the potential change, except for the two cases where Δ*V*_S,max_ is negative rather than positive. In such an event (**1**-A1→**2**-A1; **1**-A2→**2**-A2; C_pyridyl_–Ch σ-holes) ΔSF_rearrangement_ is quite small (about 0.0050 au) and counteracts the potential change.

#### 2.4.3. S→Se and R→R’ Substitutions

The double substitution case is also complex to analyze, being the result of the rearrangement of the electron distribution of the Bipy moiety and of the Ch and R group replacements. However, two clear indications may be deduced from Appendix A. First, ΔSF(Bipy) that in such case equals ΔSF_rearrangement_ by definition, is always negative and large or moderately large in magnitude. It also opposes the potential change except in one single case (**1**-A1→**5**-A1, C_pyridyl_–Ch σ-hole) where Δ*V*_S,max_ is also negative. Secondly, ΔSF(Ch) is invariably positive, quite large (0.0324−0.0886 au) and significantly overdetermining Δ*V*_S,max_, except the single case where the latter is negative. ΔSF(R) is instead always negative and smaller in magnitude than ΔSF(Ch).

#### 2.4.4. Conformational Changes

Appendix A details the last step of our Δ*V*_S,max_ analysis. Differently from Appendix A, where the ΔSF changes due to R and/or Ch replacement have been discussed keeping the same conformation type, data in Appendix A refer to the supposedly less perturbing case of a conformation change only. In this instance, we would expect (significantly) lower ΔSF values. Moreover, being by assumption ΔSF_substitution_ equal to zero, Δ*V*_S,max_ (X**→**Y) ≅ ΔSF_rearrangement_ (Equation (3)). Thus, also the ΔSF_rearrangement_ values should be small, since in this case they represent a response to only a conformational change, rather than to a substitutional change in another moiety of the molecule. For the sake of simplicity, Appendix A lists data related to the Ch σ-holes only. The Δ*V*_S,max_ values for the C_pyridyl_–Ch σ-holes are indeed small and arising, in general, from somewhat higher in magnitude ΔSF(W; W = Ch, R, Bipy) values since the Ch, R, and Bipy source differences do not have all the same sign. In practice, they may individually concur to or oppose the observed Δ*V*_S,max_ value. The potential difference values for the C_R_–Ch σ-holes are instead quite large in many cases. The composing source differences ΔSF(W) show different behaviors among themselves and an exemplar case is shown in Figure 7 (right plot). ΔSF(R) is generally quite small, analogously to the case of the C_pyridyl_–Ch holes. Thus, the ΔSF(R) values do not appreciably contribute to the σ-hole potential changes due to the conformational changes. Conversely, ΔSF(Ch) and ΔSF(Bipy) are in most cases quite large and may both concur or may, either one or the other, oppose to the observed Δ*V*_S,max_ values, according to the investigated conformational change. In summary, the quite large Δ*V*_S,max_ values for C_R_–Ch σ-holes are in many cases the result of even larger changes in the source contributions of the Ch and the Bipy moieties. These results show the noteworthy importance of the ΔSF_rearrangement_ contributions in determining hole potential changes, even when arising from the sole molecular conformational change. Indeed, the ΔSF_rearrangement_ (W) value include two contributing factors, one due to the change of the electron distribution within the W moiety and the other due to the changes in the 1/|r − r’| electron density weighting function. According to the location of the potential hole and the nature of the conformational change, the second geometrical factor may acquire a great relevance, while the former factor is generally much less important for a conformational change only. This is indeed the case for the ΔSF (W= Ch, Bipy) values of C_R_–Ch σ-holes, where large displacements (>0.2 au) of the holes from the Ch nucleus are observed upon many of the conformational changes listed in Appendix A.

## 3. Materials and Methods

The 3D structures of the configurational *M* isomers of compounds **1**–**6** were prepared using the build function, and model kits and tools provided by Spartan’ 10 Version 1.1.0 (Wavefunction Inc., Irvine, CA, USA) [48] for building and editing organic molecules. On this basis, molecules were generated, and their structure refinement was performed by a MMFF (Molecular Mechanics Force Field) procedure. Then, each structure was submitted to a conformational systematic search using the MMFF, spanning all shapes accessible to the molecule irrespective of their relative energies. After the elimination of duplicates and high-energy conformers, a set of energetically accessible conformers was selected. For each conformer, geometry optimization in vacuum was performed employing the density functional theory (DFT) method with the B3LYP functional and the 6-311G* basis set (available for elements H–Ca, Ga–Kr, and I). Computation of *V*_S_ values, given in au (electrons/bohr), was performed by using Gaussian 09 (Wallingford, CT, USA) and at DFT/B3LYP/6–311G* level [49]. Search for the exact location of *V*_S,max_ was made through the Multiwfn code [50] and through its module enabling quantitative analyses of *V*_S_ on molecular electron density isosurfaces (isovalue 0.002 au) [51]. *V*_S_ representations of all conformers of **1**–**6** depicted in the text (Figure 2 and Appendix A) were graphically generated by using Spartan’ 10 through its graphic interface. For the SF reconstruction of *V*_S,max_ values, a small code, VEXTLOC [52], was written to associate *V*_S,max_ to the various atomic basins and to select those maxima of interest for the *V*_S_ SF analysis. Atomic association was made based on the distance between each *V*_S_ extremum and each atomic nucleus in the system. Our standard SF code [53] SF_ESI, was then generalized to include the case of *V*_S_ SF reconstruction in terms of Quantum Theory of Atoms in Molecules (QTAIM) [54] basin contributions, so that both fields, electron density (ED) and *V*_S_ may be simultaneously reconstructed at the same set of reference points (rps) [55]. Such *V*_S_ SF reconstruction is quantitatively analogous to evaluating *V*_S_ as a sum of the electrostatic potential contribution of all QTAIM atomic basins of a molecule [56]. The SF_ESI code was also used to evaluate the net charges (nuclear charge minus electron population) of QTAIM atomic basins. The SF_ESI code reads wavefunction information from the .wfn files, produced as output by several ab initio packages. In our computations .wfn files were obtained through the Gaussian09 package [49]. Another small code, ANASFR_EP (Analyse SF Results for *V*_S_ field), was written to extract *V*_S_ atomic SF data calculated by the generalized SF code and to suitably combine them to get *V*_S_ SF contributions from all atomic groups defined in input [57].

## 4. Conclusions

This study focused on the interplay between halogen and chalcogen σ-holes, and aromatic π-holes located in the same class of molecules as chemical and conformational variations occur. For this purpose, conformers of six 5,5′-dibromo-2,2′-dichloro-3-chalcogen-4,4′-bipyridines were used as test probes, and *V*_S,max_ values of eight holes carried by the heteroaromatic scaffold were calculated and compared in order to evaluate the impact of chemical and conformational variations on the hole depth. With the aim to gain further insights on the contribution of atoms or groups of the molecule to generate certain *V*_S,max_ values, the Bader–Gatti SF for the electron density was extended to the electrostatic potential. Naively, a hole *V*_S,max_ is thought to increase as the remainder of the molecule becomes more electron-attracting. However, *V*_S,max_ analysis and reconstruction showed that the *V*_S,max_ values associated with σ- and π-holes located in complex molecular systems depend on the subtle balance of several effects which contribute to increase or decrease the potential value associated with the electron charge density depletion. In particular, negative or positive *V* contribution from neighboring portions of the molecule were found to significantly concur in determining *V*_S,max_ along with electronegativity and polarizability properties of bound atoms and groups. The results of this study serve the molecular design of XB, ChB, and π-hole bond donors for applicative purposes, clarifying how chemical and conformational features impact hole regions and affect their potential involvement in noncovalent interactions.

## Figures and Tables

**Figure 1 molecules-25-04409-f001:**
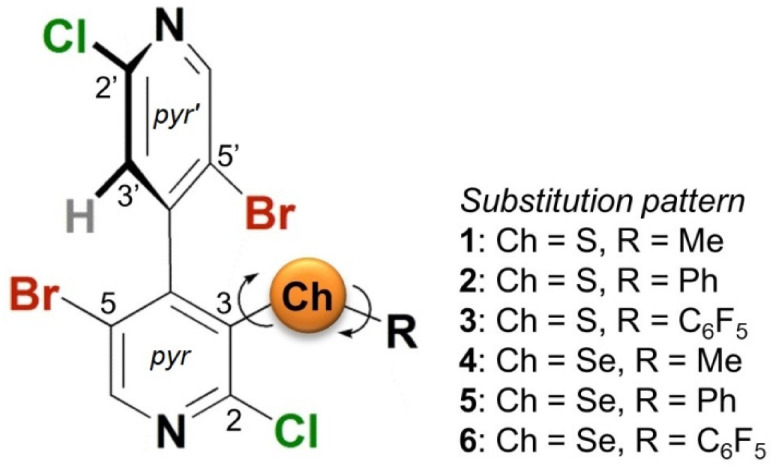
Structure of (*M*)-5,5′-dibromo-2,2′-dichloro-3-chalcogeno-4,4′-bipyridines **1**–**6**.

**Figure 2 molecules-25-04409-f002:**
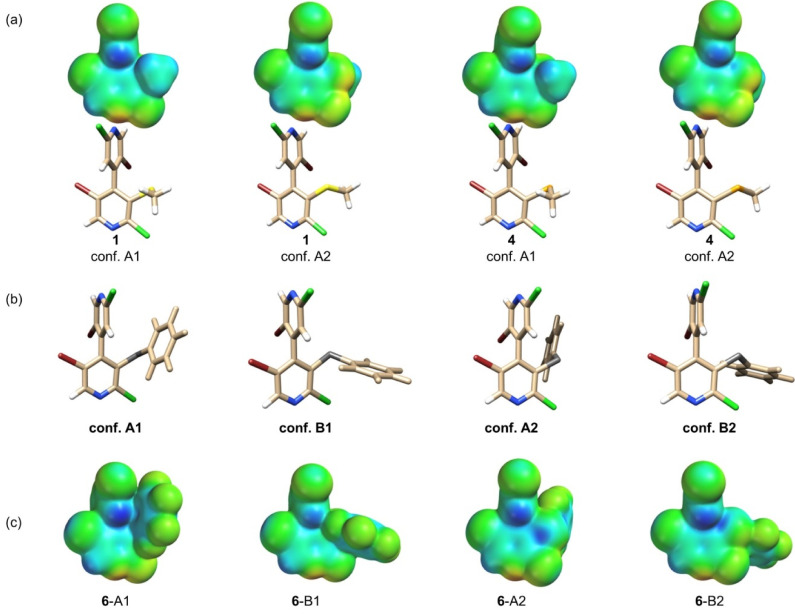
Conformations and related *V*_S_ representations on electron density isosurfaces (0.002 au) graphically generated by using Spartan’ 10 (DFT/B3LYP/6-311G*): (**a**) *V*_S_ representations and tube structures of conformers A1 and A2 calculated for compound **1** and **4**, (**b**) conformation motifs A1, B1, A2 and B2 computed for compounds **2**, **3**, and **6**, and (**c**) *V*_S_ representations of conformers A1, B1, A2, and B2 of compound **6** (*V*_S_ representations for conformers of **2**, **3**, and **5** are depicted in Appendix A). Tube structures colors: bromine (red), chalcogen S/Se (dark grey), chlorine (green), hydrogen (white), nitrogen (blue), selenium (orange), sulfur (yellow). For the *V*_S_ representations, colors towards red depict negative *V*_S_, while colors towards blue depict positive *V*_S_, and colors in between (orange, yellow, green) depict intermediate values.

**Figure 3 molecules-25-04409-f003:**
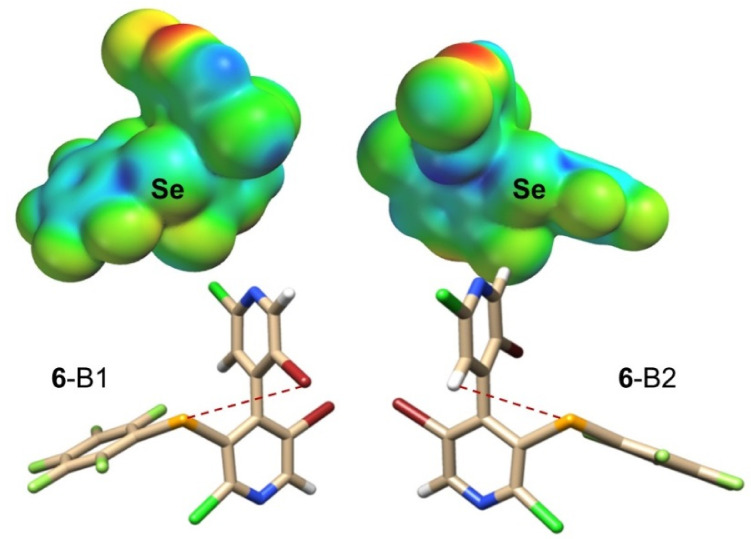
Impact of conformation environment on selenium σ-holes. Comparison of **6**-B1 and **6**-B2. Tube structures colors: bromine (red), chlorine (green), fluorine (clear green), hydrogen (white), nitrogen (blue), selenium (orange). *V*_S_ representations: colors towards red, negative *V*_S_; colors towards blue, positive *V*_S_; colors in between (orange, yellow, green), intermediate values of *V*_S_.

**Figure 4 molecules-25-04409-f004:**
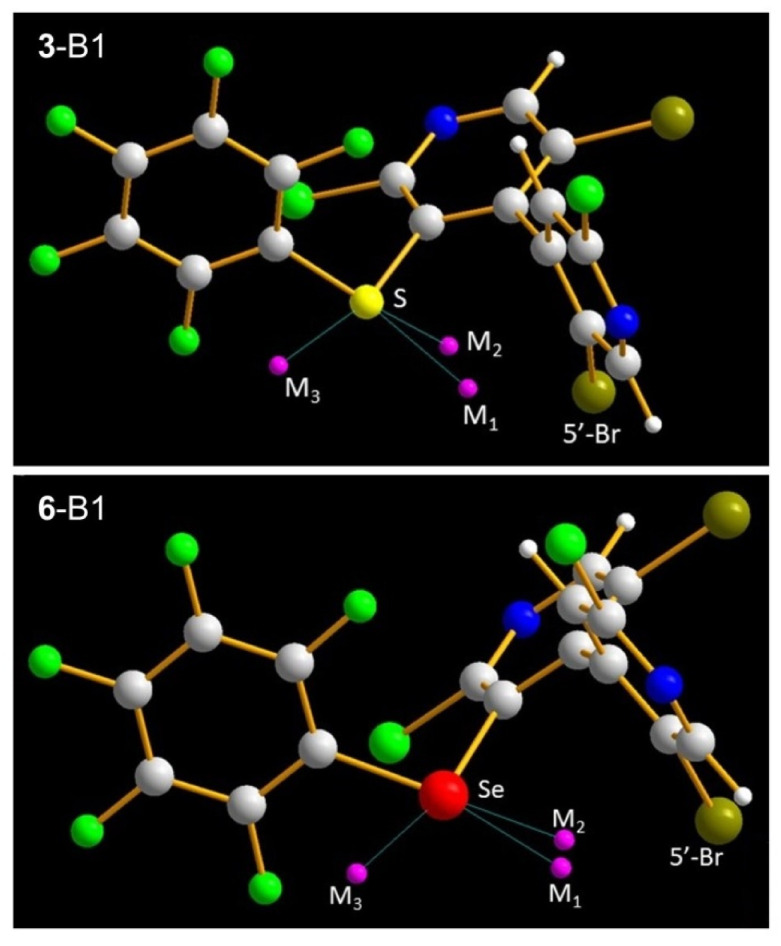
Calculated σ-holes on the elongation of C_ArF_–Ch (M1, M2) and C_Pyr_–Ch (M3) bonds in conformers **3**-B1 and **6**-B1. Tube structures colors: σ-holes (pink), bromine (dark green), carbon (grey), chlorine and fluorine (green), hydrogen (white), nitrogen (blue), selenium (red), sulfur (yellow). Pictures were obtained through the code Diamond v3.21. Putz, H. & Brandenburg, K. (1997–2012). Diamond—Crystal and Molecular Structure Visualization. Crystal Impact, GbR, Kreuzherrenstrasse 102, 53227 Bonn, Germany, http://www.crystalimpact.com/diamond.

**Figure 5 molecules-25-04409-f005:**
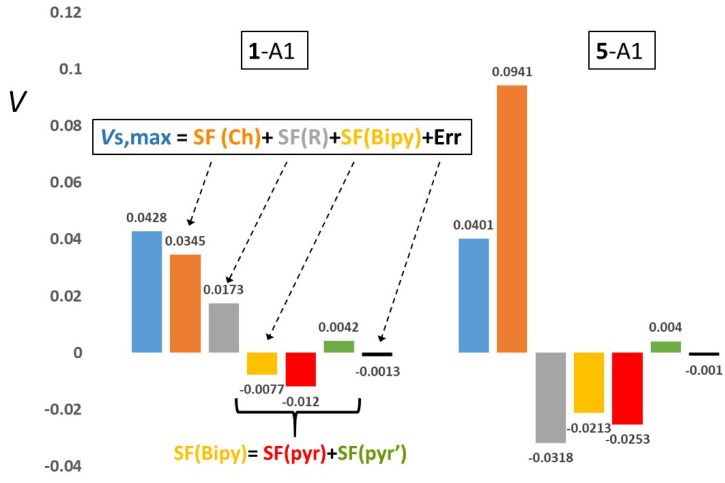
*V*_S_,_max_ (au) Source Function (SF) decomposition in two exemplar σ-hole cases (see Table 4 and Table 5 for the full list of cases and SF decomposition data). Data refer to the C_pyridyl_–Ch σ-holes (Ch = S and Se for **1**-A1 (R = Me) and **5**-A1 (R = Ph) molecules, respectively). The two molecules have very close *V*_S_,_max_ values, yet the roles of the Ch, R, and Bipy moieties in producing such values are strikingly different. The SF from the Bipy moiety, SF(Bipy), is also shown as dissected into its two composing ring contributions SF(*pyr*) and SF(*pyr’*).

**Figure 6 molecules-25-04409-f006:**
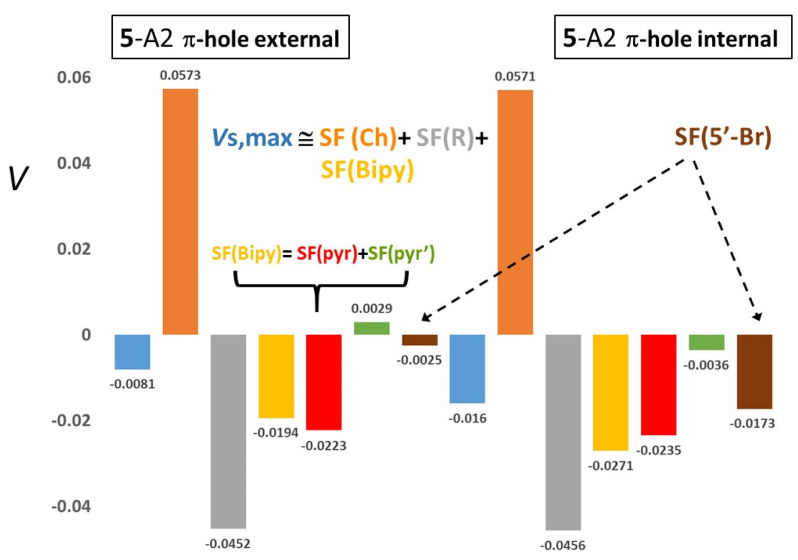
*V*_S_,_max_ (au) Source Function (SF) decomposition in two exemplar aryl π-holes (see Appendix A for the full list of cases and SF decomposition data). Data refer to the external (left) and internal (right) aryl π-holes (0.002 au molecular surface) calculated for the **5**-A2 conformer. For most of the investigated systems in this work, the internal and the external π-holes are hardly distinguishable as for their *V*_S,max_ value, but for the **5**-A2 conformer the internal π-hole has twice the *V*_S,max_ magnitude of the external one. SF contributions from the Ch = Se and R = Phenyl are almost equal for the two holes, while SF(Bipy) is the responsible for their *V*_S,max_ difference and essentially because of the different SF contribution from its *pyr*’ ring, hosting the 5′-Br atom. This atom points its π-cloud towards the phenyl ring internal π-hole making this hole largely more negative (see text).

**Figure 7 molecules-25-04409-f007:**
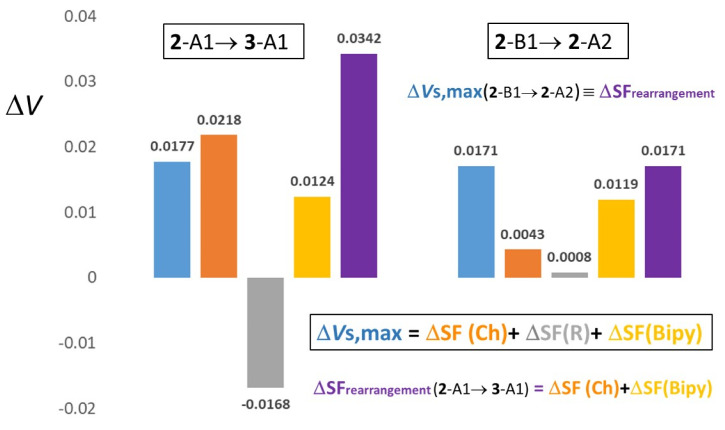
Δ*V*_S,max_ (X→Y) (au) Source Function (SF) decomposition in two exemplar cases (see Appendix A for the full list of cases and ΔSF decomposition data). Data refer to the C_R_–S σ-holes Δ*V*_S,max_ changes for a case (**2**-A1→**3**-A1) where the system undergoes chemical substitution (Ph→C_6_F_5_) and no conformational change and for a case (**2**-B1→**2**-A2) where only conformational change occurs. The two considered molecular transformations yield quite similar *V*_S,max_ variations, yet the roles of the Ch, R, and Bipy moieties in producing such similar changes strikingly differ. In particular, it is shown that a change of molecular conformation only may be as effective as a chemical substitution in its impact on the *V*_S,max_ value. The different magnitude and composition of the ΔSF_rearrangement_ term in the two cases is shown, while ΔSF_substitution_ equals ΔSF(Ch) for the **2**-A1→**3**-A1 case and it is identically equal to zero for the **2**-B1→**2**-A2 case.

**Table 1 molecules-25-04409-t001:** *V*_S,max_ (au) on halogen (Cl, Br), sulfur and selenium σ-holes (0.002 au), and the pentafluorophenyl ring π-hole calculated for conformers of compounds **1**–**6** (B3LYP/6-311G*).

Conf.	R (3-ChR)	Ch	2′-Cl	2-Cl	5′-Br	5-Br	Ch (C_pyridyl_–Ch)	Ch (C_R_–Ch)	π-Hole External ^1^	π-Hole Internal ^1^
Δ*V*_S,max_ ^2^			0.0046	0.0106	0.0078	0.0055	0.0326	0.0434	0.0597	0.0685
**1**-A1	Me	S	0.0233	0.0259	0.0449	0.0478	0.0428	0.0193		
**1**-A2			0.0219	0.0262	0.0493	0.0476	0.0423	-		
**2**-A1	Ph	S	0.0224	0.0230	0.0445	0.0475	0.0285	0.0268	−0.0080	-
**2**-A2			0.0215	0.0231	0.0451	0.0471	0.0267	0.0340	−0.0090	−0.0138
**2**-B1			0.0221	0.0222	0.0446	0.0469	0.0307	0.0169	−0.0115	−0.0087
**2**-B2			0.0215	0.0220	0.0491	0.0464	0.0286	-	−0.0127	−0.0093
**3**-A1	C_6_F_5_	S	0.0258	0.0269	0.0520	0.0492	0.0491	0.0445	0.0466	0.0538
**3**-A2			0.0246	0.0271	0.0519	0.0518	0.0491	0.0501	0.0456	-
**3**-B1			0.0235	0.0307	0.0486	0.0512	0.0495	0.0272 ^3^	0.0470	0.0444
**3**-B2			0.0250	0.0326	0.0475	0.0512	0.0512	0.0553	0.0452	0.0404
**4**-A1	Me	Se	0.0229	0.0253	0.0452	0.0471	0.0521	0.0260		
**4**-A2			0.0219	0.0256	0.0486	0.0470	0.0518	0.0463		
**5**-A1	Ph	Se	0.0225	0.0224	0.0443	0.0466	0.0401	0.0352	−0.0072	-
**5**-A2			0.0212	0.0225	0.0441	0.0463	0.0391	0.0414	−0.0082	−0.0147
**6**-A1	C_6_F_5_	Se	0.0251	0.0266	0.0493	0.0514	0.0584	0.0559	0.0454	0.0527
**6**-A2			0.0244	0.0267	0.0512	0.0512	0.0577	0.0609	0.0448	-
**6**-B1			0.0235	0.0294	0.0496	0.0510	0.0582	0.0319 ^4^	0.0455	0.0425
**6**-B2			0.0258	0.0314	0.0476	0.0510	0.0593	0.0603	0.0438	0.0390

^1^ External and internal π-holes are oriented far from and close to the 4,4′-bipyridine moiety, respectively. ^2^ Variation range of *V*_S,max_ on the corresponding σ-hole. ^3^ A third σ-hole on the elongation of the C_ArF_-S was found for the conformer **3**-B1 with *V*_S,max_ = 0.0202 au. ^4^ A third σ-hole on the elongation of the C_ArF_–Se was found for the conformer **6**-B1 with *V*_S,max_ = 0.0309 au.

**Table 2 molecules-25-04409-t002:** *V*_S,max_ (au) calculated on the S σ-holes in conformer **1**-A2 at isodensity surfaces 0.001−0.004 au.

σ-hole	0.001	0.002	0.003	0.004
σ-hole (C_pyridyl_–S)	-	0.0422	0.0502	0.0590
σ-hole (C_Me_–S)	-	-	-	0.0578

**Table 3 molecules-25-04409-t003:** *V*_S,max_ (au) and geometrical parameters of σ-holes (M) located on the elongation of C_ArF_–Ch bond in **3**-B1, **3**-B2, **6**-B1, and **6**-B2.

Conf.,M	*V*_S,max_ (au)	d_Ch-M_ (Å)	d_Br-M_ (Å)	C5′-Br-M (°)	C_ArF_-Br-M (°)
**3**-B1,M_1_	0.0202	2.098	2.268	61.9	156.2
**3**-B1,M_2_	0.0272	2.016	2.431	87.2	154.5
**3**-B2,M	0.0553	2.095	1.616 ^1^	97.6 ^2^	162.7
**6**-B1,M_1_	0.0309	2.035	2.410	73.7	163.3
**6**-B1,M_2_	0.0319	2.057	2.418	88.8	154.5
**6**-B2,M	0.0603	2.105	1.696 ^1^	94.6 ^2^	161.6

^1^ dH-M [Å]. ^2^ C3′-H-M (°).

**Table 4 molecules-25-04409-t004:** *V*_S,max_ (au) and their Source Function (SF) atomic group contributions on C_pyridyl_–Ch (Ch = S, Se) σ-holes (0.002 au molecular surface) calculated for the various conformers of systems **1**–**6**. In parentheses the SF percentage values are reported. The Err% value (see text) provides a measure of the SF *V*_S,max_ reconstruction accuracy.

Conf.	*V* _S,max_	SF(Ch)	SF(R)	SF(BiPy)	SF(*pyr*)	SF(*pyr’*)	SF(5′-Br)	SF(3′-H)	Err%
S (σ-hole, Cpyridyl–S)									
**1**-A1	0.0428	0.0345 (80.7)	0.0173 (40.4)	−0.0077 (−18.1)	−0.0120 (−28.0)	0.0042 (9.9)	−0.0016 (−3.8)	0.0101 (23.6)	3.0
**1**-A2	0.0423	0.0324 (76.5)	0.0185 (43.9)	−0.0071 (−16.8)	−0.0118 (−28.0)	0.0047 (11.2)	−0.0028 (−6.6)	0.0106 (25.1)	3.6
**2**-A1	0.0285	0.0359 (125.9)	−0.0022 (−7.6)	−0.0040 (−14.1)	−0.0088 (−31.0)	0.0048 (16.9)	−0.0017 (−5.8)	0.0115 (40.3)	4.2
**2**-B1	0.0307	0.0369 (120.2)	0.0006 (2.0)	−0.0058 (−19.1)	−0.0100 (−32.6)	0.0041 (13.5)	−0.0022 (−7.1)	0.0099 (32.3)	3.1
**2**-A2	0.0267	0.0340 (127.0)	−0.0027 (−9.9)	−0.0038 (−14.3)	−0.0095 (−35.5)	0.0057 (21.3)	−0.0000 (−0.0)	0.0118 (44.0)	2.9
**2**-B2	0.0286	0.0352 (123.1)	−0.0007 (−2.5)	−0.0056 (−19.4)	−0.0104 (−36.4)	0.0049 (17.0)	−0.0018 (−6.2)	0.0111 (38.9)	1.1
**3**-A1	0.0491	0.0656 (133.7)	−0.0197 (−40.2)	0.0042 (8.5)	−0.0023 (−4.7)	0.0065 (13.2)	−0.0010 (−2.1)	0.0107 (21.7)	2.0
**3**-B1	0.0495	0.0665 (134.3)	−0.0184 (−37.1)	0.0019 (3.8)	−0.0046 (−9.3)	0.0065 (13.1)	−0.0024 (−4.6)	0.0118 (23.7)	1.0
**3**-A2	0.0491	0.0639 (130.2)	−0.0186 (−37.9)	0.0040 (8.2)	−0.0026 (−5.3)	0.0067 (13.5)	−0.0012 (−2.5)	0.0156 (31.2)	0.5
**3**-B2	0.0512	0.0656 (128.3)	−0.0161 (−31.4)	0.0019 (3.6)	−0.0046 (−9.0)	0.0064 (12.6)	0.0009 (1.8)	0.0111 (21.6)	0.5
Se (σ-hole, Cpyridyl–Se)									
**4**-A1	0.0521	0.0897 (172.2)	−0.0122 (−23.5)	−0.0240 (−46.0)	−0.0275 (−52.7)	0.0035 (6.7)	−0.0021 (−4.1)	0.0100 (19.3)	2.7
**4**-A2	0.0518	0.0877 (169.4)	−0.0108 (−20.9)	−0.0239 (−46.3)	−0.0276 (−53.4)	0.0037 (7.1)	−0.0026 (−5.1)	0.0102 (19.8)	2.3
**5**-A1	0.0401	0.0941 (234.8)	−0.0318 (−79.2)	−0.0213 (−53.1)	−0.0253 (−63.1)	0.0040 (10.0)	−0.0018 (−4.6)	0.0112 (28.1)	2.4
**5**-A2	0.0391	0.0916 (234.2)	−0.0308 (−78.7)	−0.0210 (−53.6)	−0.0257 (−65.8)	0.0048 (12.2)	−0.0002 (−0.5)	0.0110 (28.2)	1.9
**6**-A1	0.0584	0.1227 (210.0)	−0.0495 (−84.6)	−0.0143 (−24.5)	−0.0197 (−33.7)	0.0053 (9.1)	−0.0015 (−2.5)	0.0103 (17.7)	0.8
**6**-B1	0.0582	0.1232 (211.7)	−0.0483 (−83.0)	−0.0167 (−28.6)	−0.0219 (−37.7)	0.0052 (9.0)	−0.0027 (−4.6)	0.0115 (19.7)	0.0
**6**-A2	0.0577	0.1207 (209.0)	−0.0480 (−83.2)	−0.0144 (−24.9)	−0.0201 (−34.9)	0.0057 (9.9)	−0.0015 (−2.6)	0.0111 (19.3)	0.9
**6**-B2	0.0593	0.1238 (208.7)	−0.0486 (−81.9)	−0.0160 (−27.0)	−0.0210 (−35.4)	0.0050 (8.4)	0.0007 (1.2)	0.0106 (17.9)	−0.3

**Table 5 molecules-25-04409-t005:** *V*_S,max_ (au) and their Source Function (SF) atomic group contributions on C_R_–Ch (Ch = S, Se) σ-holes (0.002 au molecular surface) calculated for the various conformers of systems **1**–**6**. In parentheses the SF percentage values are reported. The Err% value (see text) provides a measure of the SF *V*_S,max_ reconstruction accuracy.

Conf.	*V* _S,max_	SF(Ch)	SF(R)	SF(BiPy)	SF(*pyr*)	SF(*pyr’*)	SF(5′-Br)	SF(3′-H)	Err%
S (σ-hole, C_R_–S)									
**1**-A1	0.0193	0.0312 (161.1)	0.0107 (55.1)	−0.0215 (−111.1)	−0.0286(−147.6)	0.0071 (36.5)	−0.0150 (−77.4)	0.0114 (58.7)	5.1
**1**-A2	Absent								
**2**-A1	0.0268	0.0416 (155.4)	0.0038 (14.1)	−0.0175 (−65.1)	−0.0261 (−97.3)	0.0086 (32.2)	−0.0101 (−37.6)	0.0114 (42.6)	4.4
**2**-B1	0.0169	0.0342 (202.8)	0.0041 (24.6)	−0.0204 (−120.7)	−0.0263 (−155.8)	0.0059 (35.1)	−0.0166 (−98.1)	0.0115 (68.0)	6.7
**2**-A2	0.0340	0.0385 (113.2)	0.0049 (14.5)	−0.0085 (−25.0)	−0.0294 (−86.6)	0.0210 (61.7)	0.0008 (2.3)	0.0168 (49.3)	2.8
**2**-B2	Absent								
**3**-A1	0.0445	0.0634 (142.5)	−0.0130 (−29.2)	−0.0051 (−11.6)	−0.0149 (−33.4)	0.0097 (21.8)	−0.0085 (−19.2)	0.0102 (22.9)	1.8
**3**-B1	0.0272	0.0515 (189.4)	−0.0119 (−43.8)	−0.0118 (−43.5)	−0.0186 (−68.5)	0.0068 (25.0)	−0.0176 (−64.8)	0.0133 (49.0)	2.1
	0.0202	0.0267 (132.2)	−0.0121 (−59.7)	0.0062 (30.9)	−0.0004 (−2.2)	0.0067 (33.2)	−0.0064 (−31.8)	0.0153 (75.9)	3.4
**3**-A2	0.0500	0.0607 (121.2)	−0.0121 (−24.1)	0.0019 (3.8)	−0.0175 (−35.0)	0.0194 (38.8)	−0.0006 (−1.1)	0.0119 (24.3)	0.9
**3**-B2	0.0553	0.0523 (94.6)	−0.0107 (−19.4)	0.0144 (26.0)	−0.0162 (−29.4)	0.0306 (55.4)	0.0044 (7.9)	0.0262 (47.4)	1.2
Se (σ-hole, C_R_–Se)									
**4**-A1	0.0260	0.0784 (302.0)	−0.0057 (−22.1)	−0.0457 (−176.0)	−0.0495 (−190.4)	0.0037 (14.4)	−0.0167 (−64.3)	0.0115 (44.4)	3.9
**4**-A2	0.0463	0.0810 (175.0)	−0.0048 (−10.4)	−0.0290 (−62.7)	−0.0516 (−111.4)	0.0225 (48.7)	0.0011 (2.4)	0.0235 (50.8)	1.8
**5**-A1	0.0352	0.0934 (265.3)	−0.0135 (−38.3)	−0.0437 (−124.1)	−0.0501(−142.2)	0.0064 (18.1)	−0.0104 (−29.6)	0.0113 (32.2)	2.9
**5**-A2	0.0414	0.0899 (216.9)	−0.0124 (−29.9)	−0.0352 (−84.9)	−0.0532 (−128.3)	0.0180 (43.4)	0.0006 (1.6)	0.0158 (38.1)	2.0
**6**-A1	0.0559	0.1212 (216.9)	−0.0301 (−53.9)	−0.0346 (−62.0)	−0.0421(−75.3)	0.0074 (13.3)	−0.0093 (-16.7)	0.0101 (18.1)	1.0
**6**-B1	0.0309	0.0839 (271.6)	−0.0286 (−92.6)	−0.0242 (−78.3)	−0.0244 (−79.0)	0.0002 (0.7)	−0.0147 (−47.5)	0.0137 (44.3)	0.7
	0.0319	0.1011 (317.0)	−0.0288 (−90.4)	−0.0400 (−125.6)	−0.0425 (−133.4)	0.0025 (7.7)	−0.0195 (−61.3)	0.0130 (40.6)	0.9
**6**-A2	0.0609	0.1176 (193.1)	−0.0288 (−47.4)	−0.0272 (−44.7)	−0.0444 (−72.9)	0.0172 (28.2)	−0.0007 (−1.2)	0.0149 (24.5)	1.0
**6**-B2	0.0603	0.1059 (175.5)	−0.0284 (−47.1)	−0.0161 (−26.8)	−0.0391 (−64.9)	0.0230 (38.1)	0.0043 (7.2)	0.0254 (42.1)	1.7

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
