# Peer review of "Factors Impacting σ- and π-Hole Regions as Revealed by the Electrostatic Potential and Its Source Function Reconstruction: The Case of 4,4′-Bipyridine Derivatives"

_molecules, 2020, doi:10.3390/molecules25194409_

Round 1
Reviewer 1 Report
The study titled: “Factors impacting s- and p- hole regions as revealed by the electrostatic potential and its source function reconstruction: the case of 4,4’-bipyridine derivatives” from my point of view is interesting, since a systematic work is done with a 4,4’-bipyridine family compounds. A clear objective is addressed: the study of V values on electron density isosurfaces (Vs) dissecting the atomic group contributions the Vs maxima. Technically correct with the necessary quality parameters for precise comparison, with a good number of comparisons and different points of view. The paper is correctly structured and the conclusions obtained are clear. For me it can be published in its current state.
Author Response
We thank the referee for her/his positive comments on our work and for the time she/he spent for revising it.
Reviewer 2 Report
This is excellent work, using DFT and electron density/electrostatic potential to analyse the halogen, chalcogen and aromatic "holes" in a set of model compounds that vary in subsitution pattern. It exemplifies the variation in number and depth of such holes very clearly as a function of conformation and chemical substitution. This is already interesting, but the use of the Bader-Gatti source function deepens the analysis considerably and shows how different groups within the structure act to enhance or reduce the various holes.
This should be of considerable interest to the readership of this journal, and I am pleased to recommend publication. My only comments for revision is that it is sometimes hard work to read, with long wordy paragraphs and very large tables. I would encourage the authors to consider moving tables to ESI and making more use of images such as Figure 5, which neatly encapsulate the important aspects of the data generated.
Author Response
We thank the referee for her/his positive comments on our manuscript and for the time she/he spent for revising it.
We have also carefully followed the useful suggestions she/he provided.
In particular:
In order to decrease the information conveyed through Tables and increase that immediately explained through Figures, we have changed our manuscript as follows:
a) Table 6 (SF decomposition of VS,max potentials of the external and internal π-holes) has been removed from the main text and it has been placed in the SI (Table S5).
b) We have added a new Figure (Figure 6) in the same spirit of Figures 5-6 of the old manuscript (which have been judged favourably by the referee). The new figure 6 illustrates one of the more interesting results obtained by the SF decomposition of the VS,max potential of external and internal π-holes.
c) we thought it important to keep in the text Tables 4 and 5, since the Source Function decomposition of the sigma-holes potentials (detailed in those Tables) is one of the most relevant results of our work. It would have been difficut for a reader to follow our discussion, having those Tables only in the SI. Figures 5 and 6 of the old manuscript version (5 and 7 in the new version) and now also Figure 6 are surely immediate, but represent just few exemplar cases. We feel taht it is the combination of these Figures and of Tables 4-5 and S5-S7 that enable the reader to catch the full spirit of our work.
Finally, we have changed a number of sentences throughout the text that, despite being correct and surely understandable, were actually a bit too long. These sentences have been each suitably broken in two new sentences, making them more readable and without changing their meaning in any way.